# Atomic Magnetometer Achieves Visual Salience Analysis in Drosophila

**DOI:** 10.3390/s23031092

**Published:** 2023-01-17

**Authors:** Fan Liu, Dongmei Li, Yixiao Li, Zhao Xiang, Yuhai Chen, Zhenyuan Xu, Qiang Lin, Yi Ruan

**Affiliations:** Zhejiang Provincial Key Laboratory, and Collaborative Innovation Center for Quantum Precision Measurement, College of Science, Zhejiang University of Technology, Hangzhou 310023, China

**Keywords:** atomic magnetometer, visual salience, short-term memory

## Abstract

An atomic magnetometer (AM) was used to non-invasively detect the tiny magnetic field generated by the brain of a single Drosophila. Combined with a visual stimulus system, the AM was used to study the relationship between visual salience and oscillatory activity of the Drosophila brain by analyzing changes in the magnetic field. Oscillatory activity of Drosophila in the 1–20 Hz frequency band was measured with a sensitivity of 20 fT/$\sqrt{\mathrm{Hz}}$. The field in the 20–30 Hz band under periodic light stimulation was used to explore the correlation between short-term memory and visual salience. Our method opens a new path to a more flexible method for the investigation of brain activity in Drosophila and other small insects.

## 1. Introduction

Drosophila is capable of complex cognitive functions such as associative learning [1], novel learning [2] and contextual generalization [3]. The key point of these abilities lies in the assignment of salience to stimuli [4]. From behavioral observations only, it is difficult to distinguish between the ability of salience assignment and the ability to perform motor tasks at hand [5], so that electroencephalogram (EEG) and local field potential (LFP) are implemented for investigations [6]. Most of the complex cognitive functions in Drosophila involve vision. Bruno van Swinderen et al. studied the brain responses evoked by visual stimuli and the modulation of these responses in the 20–30 Hz range by salience. The attention-like process with stereotyped temporal characteristics was revealed by the recording of the local field potential of a visual novelty response [4,7]. Amber McCartney et al. uncovered genetic and neuroanatomical systems in the fly brain affecting both visual attention and odor memory phenotypes. A common component to these systems appeared to be the mushroom bodies, brain structures which have been traditionally associated with odor learning but which might also be involved in generating oscillatory brain activity required for attention-like processes in the fly brain [8]. Memory, on the other hand, provides a mechanism to detect stimulus change; it can suppress or enhance 20–30 Hz oscillatory activity in response to visual stimuli, which was confirmed in the fixation experiment of memory mutants dunce and rutabaga [9]. This was also confirmed in the work of Zhihua Wu et al.—the selective visual attention behavior of Drosophila was tested by employing the flight simulator. It has been found that the learning memory mutants dunce and amnesiac possess attention patterns totally different from that of the wild-type fly [10]. These studies have demonstrated that visual salience is closely related to the 20–30 Hz response of Drosophila, and the regulation of mushroom-body and short-term memory genes on it. However, EEG and LFP are invasive, manually complex and localized to the brain, and are limited to the detection of the whole brain. Moreover, although MEG has been intensively used [11,12], for insect brains, which are exceedingly small, a great challenge is presented by the attempt to attain higher spatial resolution and sensitivity.

High-sensitivity magnetometers, such as fluxgate magnetometers, superconducting quantum interference devices (SQUIDs) and atomic magnetometers (AMs), can be used to non-invasively detect the tiny magnetic fields in the brain [13,14,15,16]. However, the sensitivity of the fluxgate magnetometer is ∼30 fT/$\sqrt{\mathrm{Hz}}$ [17]. The main drawback of the SQUID magnetometer is the requirement of cryogenic cooling to maintain its high performance, making the costs for installation and maintenance high [18,19]. AM, based on atom–light interaction, has shown extremely high sensitivity—no worse than SQUID—and it has no requirement for cryogenics [20,21,22,23]. Therefore, to detect the tiny magnetic field signal of Drosophila, we used a spin-exchange relaxation-free atomic magnetometer (SERF AM), which only worked with no environmental magnetic field (<10 nT), to detect the Drosophila brain responses evoked by visual stimuli. Furthermore, we revealed the correction between short-term memory and visual salience, which may provide novel perspectives on visual memory, visual learning, decision making, etc.

Statement: All animals were kept in a pathogen-free environment and fed ad lib. The procedures for care and use of animals were approved by the Ethics Committee of the Zhejiang University of Technology, and all applicable institutional and governmental regulations concerning the ethical use of animals were followed.

Animals: Drosophila used in the experiments were wild-type red-eyed Drosophila melanogaster, it they were cultured on a diet consisting of yeast, corn syrup and agar at 25 ${}^{\circ }$C. Drosophila were stationary; there were only occasional spontaneous movements of their feet during the experiment.

## 2. SERF AM

The internal structure of SERF AM probe is shown in Figure 1A. A 3×3×3 mm gas cell containing a drop of enriched ${}^{87}$Rb atoms and 760 Torr of $N_{2}$ gas was placed on the front of the probe. It was heated to 160 ${}^{\circ }$C in the experiment via a 1300 kHz AC heating current. Three mutually orthogonal Helmholtz coils were wound outside the gas cell to cancel the residual magnetic field by using a DC current source for power supply, which can greatly minimize noise. The compensation currents were automatically determined by a program, which detects the residual magnetic field by measuring the atomic signal. The magnetic field generated by the coils can range from −30 to 30 nT. A 10 Hz oscillating magnetic field signal was added to the *z* axis, and the current of the three-axis coil was adjusted to reach its maximum amplitude, and the influence of the background magnetic field on the sensitivity was adjusted to the best level. The light was tuned to blue shift of dozens of GHz with respect to the ${}^{87}$Rb $D_{1}$ transition $F=2\to F^{{}^{\prime }}=1$. Before the light passes through the atomic gas chamber along the *x* axis, the beam is expanded by a plano-convex lens, so that more atoms participate in the interaction between light and atoms and improve the signal to noise ratio (SNR). Then, its polarization ellipticity is adjusted by changing the relative angle $\theta_{0}$ between the optic axes of the linear polarizer and the quarter-waveplate. In our experiment, we set $\theta_{0}=\frac{\pi }{8}$, which gives a 45${}^{\circ }$ elliptically polarized laser beam. The elliptical polarization state of light can be regarded as a superposition of linear polarization state and circular polarization state. The circular component of the light creates relatively uniform spin polarization, and the linear component is used to measure optical rotation generated by the atoms. The following is the Bloch equation describing the polarization state of atoms in the cell [24,25,26,27,28]:(1)$$\frac{\partial P}{\partial t}=\frac{1}{Q\left(\left|P\right|\right)}\left(\gamma_{e}P\times B+R_{OP}\left(s\hat{x}-P\right)-R_{rel}P\right)$$
where P is an atomic polarization state. $Q\left(\left|P\right|\right)$ is the slowing-down factor. The first term is the precession of the atomic polarizability under the action of the external magnetic field B, and $\gamma_{e}$ is the electron gyromagnetic ratio. The second term is the relaxation rate brought by the optical pump, *s* is the photon polarization degree of the pump light and $R_{OP}$ is the rate of absorption of pump light by atoms. The last term, $R_{rel}$, is the relaxation rate caused by factors other than the pump light. In order to measure the magnetic field $B_{z}$ in the direction *z*, we used the modulated magnetic field $B_{mod}=B_{mod}cos\omega t\hat{z}$ to transfer the signal to a higher frequency ω and input the recorded signal into the lock-in amplifier to demodulate the magnetic field information. When the modulation parameters are optimized, $\omega \gg R+\frac{1}{T_{2}}$, where $T_{2}$ is the transverse relaxation time of ${}^{87}\mathrm{Rb}$; the atomic magnetization vector in the *x* direction is [29]
(2)$$P_{x}=\frac{2sR\tau B_{z}\tau^{2}sin\omega t}{1+\gamma^{2}B_{z}^{2}\tau^{2}}J_{0}\left(\frac{\gamma B_{mod}}{q(P)\omega }\right)J_{1}\left(\frac{\gamma B_{mod}}{q(P)\omega }\right)$$
where P is the atomic polarization, γ is the electron gyromagnetic ratio, q(P) is the nuclear slowing down factor that depends on the spin polarization, $\tau =R+\frac{1}{T_{2}}$, s is the average angular momentum of the incident photons and $J_{0},J_{1}$ are Bessel functions of the first kind. The magnetic field $B_{z}$ is determined by the optical rotation of laser polarization after passing through the cell. The rotation angle can be calculated by the following equation:(3)$$\theta =\frac{\pi }{2}\ln r_{e}cP_{x}\left(-f_{D_{1}}Im\left[V\left(v-v_{D_{1}}\right)\right]+\frac{1}{2}f_{D_{2}}Im\left[V\left(v-v_{D_{2}}\right)\right]\right)$$
where $v_{D_{1}}$ and $v_{D_{2}}$ are the resonance frequencies of the $D_{1}$ and $D_{2}$ transitions, respectively. *l* is atomic cell length; n is the atomic number density of ${}^{87}\mathrm{Rb}$; $r_{e}$ is atomic radius of ${}^{87}\mathrm{Rb}$; $P_{x}$ is polarization intensity in *x* direction; $f_{D_{1}}$ and $f_{D_{2}}$ are the oscillator strength. In our work, the optical rotation was measured by a balanced polarimeter, which consists of a half-wave plate, a Wollaston prism and a balanced photodetector. More details on the calculation between the optical rotation and the magnetic field can be found in reference [28,30]. After optimization, the sensitivity of SERF AM was 20 fT/$\sqrt{\mathrm{Hz}}$, and the bandwidth was 1 to 80 Hz (Appendix A).

In Figure 1B, Drosophila were glued to a silica aerogel pad, and the pad was then placed it on surface of ${\mathrm{AM}}_{2}$, making it at the center of the air chamber—that is, 4 mm from the top, which was followed by ${\mathrm{AM}}_{1}$ with a distance of 5 cm in the same direction. Since the signal detected by the AM attenuates as distance increases, the effect of Drosophila’s magnetic field on ${\mathrm{AM}}_{1}$ can be ignored. Thus, the total magnetic field from Drosophila and the environment could be collected by ${\mathrm{AM}}_{2}$. Exclusively, ${\mathrm{AM}}_{1}$ and ${\mathrm{AM}}_{2}$ were both in the same shielding room, and the environmental noise levels collected by ${\mathrm{AM}}_{1}$ and ${\mathrm{AM}}_{2}$ were almost the same, so the environmental noise can be eliminated by subtracting the data recorded by ${\mathrm{AM}}_{1}$ from the data recorded by ${\mathrm{AM}}_{2}$. Therefore, in our work, two SERF AMs were developed to minimize the background noise. Notice that silica aerogel pad between ${\mathrm{AM}}_{2}$ and Drosophila was used for thermal insulation to avoid a fatal injury to Drosophila caused by a 45–50 ${}^{\circ }$C surface temperature of AM. Moreover, it is shown that the temperature will affect the contour and peak position of response in the frequency band [31,32]. With the 2 mm silica aerogel pad, the surface temperature was reduced to 25–30 ${}^{\circ }$C. In addition, it was proved that 510 nm green light produces the greatest signal in Drosophila [33,34]. Therefore, in our experiments, light stimulation was introduced into the shield through a green LED light with a central wavelength of 510 nm by an optical fiber. The light spot was shaped into a vertical strip light and applied directly to the Drosophila by a vertical grating.

## 3. Results

Signals from Drosophila in the absence of stimulation, continuous light stimulation, and those from dead flies were recorded and spectrally analyzed. Data represented in Figure 2A were obtained by using the periodogram method to have power spectral density analysis of the recorded data. The ordinate represents the power spectral density (PSD), and the abscissa shows the frequency. Figure 2B is frequency response curve of AM. Figure 2A,B show that the PSD in the entire frequency range from 1 to 20 Hz was significantly increased under “light on” compared with “light off”. This phenomenon fully demonstrated the magnetic field detection feasibility of AM in Drosophila. The PSD at 1–20 Hz was higher in the absence of light than that of the dead Drosophila, which may because the Drosophila has a response to other unknown environmental factors, such as temperature, odor, vibration and even uncontrolled movements of itself (control measurements, Appendix A).

Next, in order to study the regulation of visual saliency on the response of the Drosophila brain at 20–30 Hz, the continuous light stimulation was changed to a period of 4 s with the light on for 1 s and the light off for 3 s. Notice that the vertical-stripe light is more significant for Drosophila’s vision than other shapes of light [35,36]. A more significant 20–30 Hz response was stimulated [4]. In the absence of the stimulus, we set the recording of data to the same time interval as that for the stimulus and performed the analysis. As the absolute power of different Drosophila varied considerably, we introduced an indicator here to measure the power variation at 20–30 Hz, with a normalized change metric: NCM [4,37]. It was a normalized ratio metric used to describe the pre-post difference in visual saliency. Firstly, the data were divided into time bins of relevant size according to the stimulus duration shown in Figure 3A. For example, the data recorded in the period of no stimulation A form a tome bin, and the power at the 20–30 Hz frequency band of each bin was calculated as $I=A/\bar{T}$, where A is the 20–30 Hz power in the bin being analyzed, and $\bar{T}$ is the average of the 20–30 Hz power of all bins required in the figure. For example, in the first cycle, there was NCM of 20–30 Hz during light stimulation:$${\mathrm{NCM}}_{A_{1}}=A_{1}/\left(\left(A_{1}+\cdots +A_{5}+B_{1}+\cdots +B_{5}\right)/10\right)$$

$A_{1}-A_{5}$ are the power levels of 20–30 Hz when there was light in the first five stimulation cycles, respectively. $B_{1}-B_{5}$ are the power of 20–30 Hz when there is no light in the first five stimulation cycles. The electronic noise mainly came from the signal source and current source used to control the led lamp. No stimulation A was the non-open signal source and current source scenario, and no stimulation B was the open signal source and current source scenario. From Figure 3B, it can be found that the 20–30 Hz response in the absence of stimulation did not change significantly: *p* > 0.05, n = 11 flies, and each fly was recorded forthe first five stimulation cycles. It was demonstrated that the electronic noise could be excluded by the NCM. As indicated in Figure 3C, the 20–30 Hz response in the presence of stimulation was significantly increased compared with that without stimulation: *p* < 0.05; n = 11 flies. However, the NCM of other bands had no significant change compared without stimulation; *p* > 0.05 and n = 11 flies (Appendix A). Each fly was recorded in the first five stimulation cycles, which indicated that the 20–30 Hz response was affected by the regulation of visual salience. Data from 11 flies recorded in Figure 3D showed that the 20–30 Hz response for cycles 1–10 was significantly decreased compared to the 20–30 Hz response for cycles 1–5; *p* < 0.05. There was also a significant increase in 1–10 cycles compared to the no-stimulus 20–30 Hz response; *p* < 0.05.

For a further exploration about the 20–30 Hz response signal over time, we analyzed every two cycles for 1–12 cycles. As shown in Figure 4, the 20–30 Hz response is significantly higher in the first four cycles compared with that without stimulation, whereas the fifth cycle decreased to the level of no stimulation and then stabilized. Figure 4 records data from 11 flies. The reason for the drop in response at 20–30 Hz may be explained by the Drosophila adapting to this periodic light stimulation after four cycles of periodic light stimulation.

In summary, a 1–20 Hz signal from Drosophila in continuous light stimulation was detected for demonstration of the feasibility of AM. Then, we confirmed the correlation between visual salience and a 20–30 Hz response in the Drosophila brain. In addition, short-term memory that was associated with a decline in 20–30 Hz response was analyzed. We found that after four cycles of light stimulation, the 20–30 Hz response dropped to almost the same level as that of the situation with no stimulation, and it may have resulted from the fact that salience requires short-term memory to discriminate. Salience is a marker of change, and memory is the basis for detecting the existence of change. Formation of stimulus short-term memory enhanced or inhibited salience; therefore, the 20–30 Hz response can also be enhanced or reduced, which correlated with visual salience. These results provide proof that short-term memory could be formed by after visual stimuli in Drosophila, which in turn inhibits the 20–30 Hz response.

In this work, a SERF AM for detection of the Drosophila brain’s brain magnetic field was developed, which has the merits of non-invasiveness and easy manipulation. Our results echoed the study of Drosophila by using EEG and LFP; moreover, we revealed the correlation between short-term memory and visual salience. It also could be used for subsequent research on mutant-Drosophila-associated short-term memory genes and mushroom bodies. Our method opens a novel path to a more flexible investigation method of the brain activity in Drosophila and other insects that are small.

## Figures and Tables

**Figure 1 sensors-23-01092-f001:**
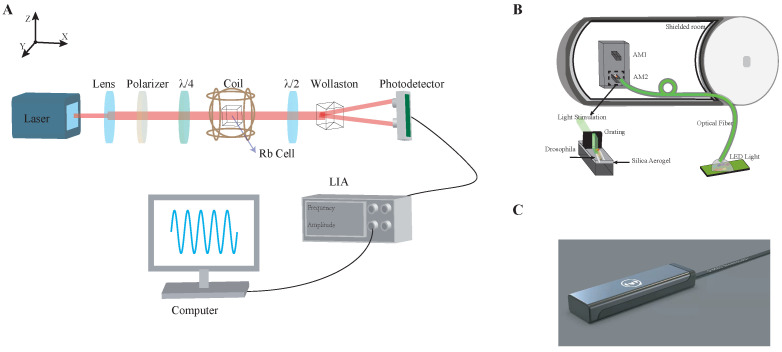
Configuration of the setup: (**A**) Internal structure diagram of SERF AM. (**B**) A five-layer magnetic shield was employed. Both the grating and the silica aerogel pad are made of non-magnetic materials. The thickness of the silica aerogel pad is 2 mm. During the measurement, ${\mathrm{AM}}_{1}$ and ${\mathrm{AM}}_{2}$ and Drosophila were placed in the shield, and the measurement direction was the *z* axis. (**C**) The SERF AM. The shell is made of high-temperature-resistant and non-magnetic nylon material. The center of the cell is 4 mm away from the shell. The figure has been authorized by Hangzhou Q-Mag Technology Co., Ltd., Hangzhou, China.

**Figure 2 sensors-23-01092-f002:**
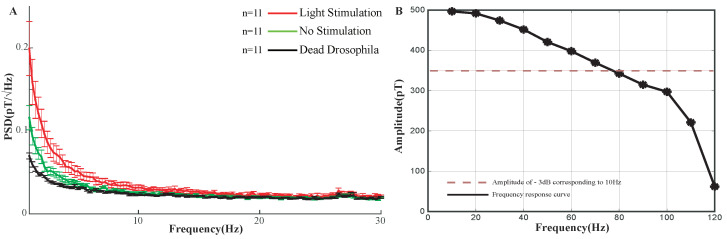
(**A**) Response to continuous light stimulation. Power spectral density (PSD) between 1 and 30 Hz in dead flies (black line) and live flies in the unstimulated state (green line), and in live flies under light stimulation (red line). Each line represents the mean and standard deviation calculated from measurements of multiple flies. Number of measurements per condition: N = 11 (dead flies, black line), N = 11 (live flies without stimulation, green line), N = 11 (live flies with light stimulation, red line). (**B**) Frequency response curve with the amplitude beginning at 500 pT.

**Figure 3 sensors-23-01092-f003:**
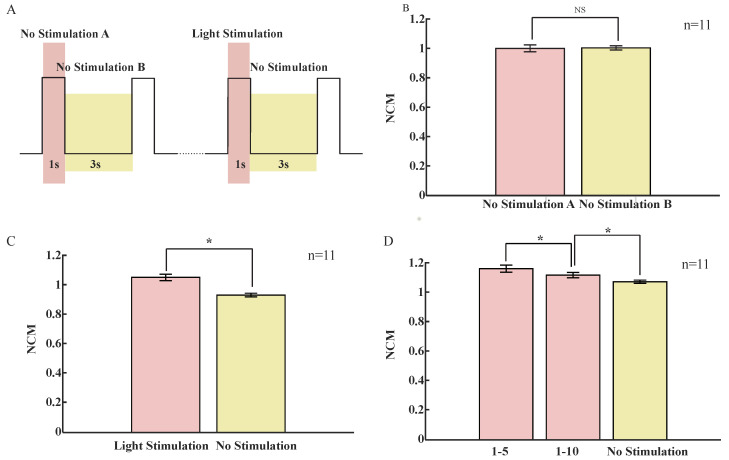
Response to periodic light stimulation. (**A**) Diagram illustrating the stimulus. Flies were subjected to 1 s of light stimulation at 3 s intervals. (**B**) Average 20–30 Hz NCM (±SEM) during no stimulation A (yellow) and no stimulation B (red); n = 11 flies; the first five cycles were recorded. (**C**) Average 20–30 Hz NCM (±SEM) during no stimulation (yellow) and light stimulation (red); n = 11 flies; the first five cycles were recorded. (**D**) Average 20–30 Hz NCM (±SEM) for cycles 1–5 (red) and 1–10 (red) and average 20–30 Hz NCM (±SEM) during the no-stimulus period (yellow) in the first 10 cycles; n = 11 flies. Significance was defined by two-tailed *t*-test; *, *p* < 0.05; NS, *p* > 0.05.

**Figure 4 sensors-23-01092-f004:**
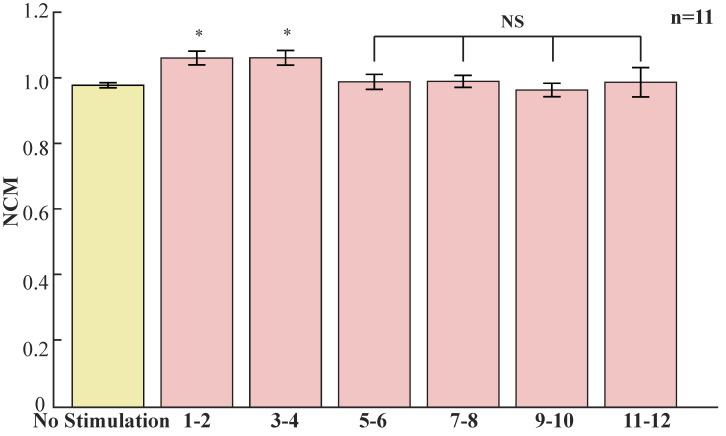
Continuous light stimulation reduced the 20–30 Hz response. No stimulation (yellow) compared to 1–2, 3–4, 5–6, 7–8, 9–10 and 11–12 cycles during light exposure (red), on average, 20–30 Hz NCM (±SEM); n = 11 flies; the first twelve cycles were recorded. Significance was determined by two-tailed *t*-test; *, *p* < 0.05; NS, *p* > 0.05.

## Data Availability

The data presented in this study are available on request from the corresponding author.

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
