# Peer review of "Atomic Magnetometer Achieves Visual Salience Analysis in Drosophila"

_sensors, 2023, doi:10.3390/s23031092_

Round 1
Reviewer 1 Report
I recommend doing a control experiment when a signal is detected from other parts of the fly and not just the brain. It can be assumed that the static magnetic properties of the fly are the source of the signal
Reviewer 2 Report
The atomic magnetometer was used to detect the Drosophila brain magnetic field and analyze the correlation between visual saliency and Drosophila brain activity. The study is interesting and novel. However, I think there are some possible errors and the content needs to be greatly supplemented:
1. In Section 2, the authors claim that the frequency of electric heating of the cell is introduced as 1300 Hz whereas in other studies it is usually tens to hundreds of kHz. The higher the modulation frequency, the smaller the impact on the atoms. Please explain why the heater frequency in this study is much lower than that of the conventional experiments.
2. The "magnetic field strength" is used many times in the manuscript to indicate the size of the magnetic field with the unit of nT. However, the magnetic field strength is H in A/m. It should be expressed as magnetic induction, or magnetic flux density, or simply as magnetic field instead.
3. In Figure 1, the author claims that the atomic magnetometer used is "homemade", but also mentions “from Hangzhou Q-Mag Technology Co. Ltd”. This is contradictory.
4. In the part of introducing the principle of atomic magnetometer, the atomic magnetometer used is introduced as using elliptically polarized light, and the measured angle of polarization is given by formula (7). However, the expression of formula (7) is wrong, θ should be Px/P0, not Pz/P0. In addition, for the atomic magnetometer using elliptically polarized light, it is necessary to apply a transversely modulated magnetic field to determine the measurement axis, otherwise the magnetometer cannot determine whether the measurement is in the x direction or the y direction. Here the author needs to carefully confirm the specific measurement method, and give an accurate description and expression in the manuscript. Reference: V. Shah and M. Romalis. Spin-exchange relaxation-free magnetometry using elliptically polarized light, Physical Review A, 2009.
5. At the end of page 3 of the manuscript, it is mentioned that " AM1 was here for the collection of the environmental noise, which could be removed by subtracting the data recorded by AM1 from the data recorded by AM2". AM1 and AM2 are placed in different positions in the magnetic shield. I wonder if this is an accurate estimate of environmental noise at AM2. Alternatively, could you please provide a discussion of environmental noise within the magnetic shield?
6. Due to the small size of Drosophila, the difference in the sticking position relative to AM2 will affect the significance of the magnetic field measurement, please give a specific discussion on the sticking position of Drosophila.
7. In the first part of the results, when the brain activity of a single Drosophila was measured with light stimulation and no stimulation, whether the order of the two in time would affect the measurement results. That is, measurements with light stimulation first and then no stimulation, and measurements with no stimulation first and then light stimulation.
8. As shown in Figure 2, PSD is used in this study to analyze the signals from Drosophila. AM has different response capabilities for each frequency. I suggest giving the frequency response curve of the AM used under the same conditions, or directly processing the analyzed PSD signal considering the frequency response, which may give more meaningful results.
9. In Figure 3 and Figure 4, the number of scales of the vertical axis should be increased.
10. The second paragraph on page 2 mentions “It was demonstrated that the electronic noise could be excluded by the NCM.” Please explain further on this.
11. As shown in Figure 4, the response in the 11-12 cycle is higher than that in the 9-10 cycle, please give a reasonable explanation for this phenomenon.
12. The references in the manuscript are out of order and need to be recited in the proper place.
Reviewer 3 Report
Dear Authors,
Congratulation on the interesting paper. I enjoy reading it. The paper is well-structured and organized.
I only have found a few typos and inconsistencies in references, which must be corrected.
Typos:
-
In the second term in Eq. (1), a unit vector along the direction of light propagation (z-axis) is missing, it should be ROP(sz – P), see for example Eq. (1) in Ref. [27] or Eq. (4) in Ref. [29]
-
The nominator in Eq. (7) should be Px instead of Pz
References:
-
References [23] and [36] are the same.
-
In the last sentence of the first paragraph in the Introduction section you wrote: “EEG has been intensively used [11,12]”, but these two references are about using MEG, not EEG. You should either refer to proper reference(s) or rephrase the whole sentence.
Round 2
Reviewer 2 Report
The authors have made corrections according to all my comments. Therefore, I recommended the manuscript should be accepted.